# Exploring the Gamut of Receptor Tyrosine Kinases for Their Promise in the Management of Non-Alcoholic Fatty Liver Disease

**DOI:** 10.3390/biomedicines9121776

**Published:** 2021-11-26

**Authors:** Sayali Bhave, Han Kiat Ho

**Affiliations:** Department of Pharmacy, Faculty of Science, National University of Singapore, 18 Science Drive 4, Singapore 117559, Singapore; sayali_bhave@u.nus.edu

**Keywords:** non-alcoholic fatty liver diseases, receptor tyrosine kinases, steatosis, fibrosis, EGFR, c-MET, AXL, FGFR, VEGFR

## Abstract

Recently, non-alcoholic fatty liver disease (NAFLD) has emerged as a predominant health concern affecting approximately a quarter of the world’s population. NAFLD is a spectrum of liver ailments arising from nascent lipid accumulation and leading to inflammation, fibrosis or even carcinogenesis. Despite its prevalence and severity, no targeted pharmacological intervention is approved to date. Thus, it is imperative to identify suitable drug targets critical to the development and progression of NAFLD. In this quest, a ray of hope is nestled within a group of proteins, receptor tyrosine kinases (RTKs), as targets to contain or even reverse NAFLD. RTKs control numerous vital biological processes and their selective expression and activity in specific diseases have rendered them useful as drug targets. In this review, we discuss the recent advancements in characterizing the role of RTKs in NAFLD progression and qualify their suitability as pharmacological targets. Available data suggests inhibition of Epidermal Growth Factor Receptor, AXL, Fibroblast Growth Factor Receptor 4 and Vascular Endothelial Growth Factor Receptor, and activation of cellular mesenchymal-epithelial transition factor and Fibroblast Growth Factor Receptor 1 could pave the way for novel NAFLD therapeutics. Thus, it is important to characterize these RTKs for target validation and proof-of-concept through clinical trials.

## 1. Introduction

Non-Alcoholic Fatty Liver Disease (NAFLD) is an umbrella term which encompasses various liver ailments arising from simple steatosis, and potentially advancing into chronic conditions of non-alcoholic steatohepatitis (NASH), liver fibrosis (with varying grades of severity) and cirrhosis [1]. Currently, it is one of the most prevalent diseases, affecting approximately 25% of the world’s population [2,3]. More critically, NAFLD is also associated with the development of hepatocellular carcinoma (HCC) which is the 6th most common cancer and 3rd largest cause of cancer related deaths. NAFLD is also associated with increased risk of cardiovascular diseases and chronic kidney diseases [4,5]. Recent reports also claim that NAFLD increases the risk and severity of COVID-19 infection [6,7]. This puts NAFLD at the forefront of public health concerns, necessitating early intervention to save lives and reduce the burden on healthcare systems.

NAFLD could arise from various triggers such as metabolic, genetic, or environmental conditions. NAFLD has been strongly associated with metabolic syndromes such as type II diabetes, where insulin resistance is one of the prime factors in pathogenesis of NAFLD. Likewise, obesity is also considered as major risk factor for NAFLD. A plethora of studies confirm the association of various genetic variants to the development of NAFLD. For instance, Single Nucleotide Polymorphisms (SNPs) in the gene encoding patatin-like phospholipase domain-containing protein 3 (PNPLA3) [8], transmembrane 6 superfamily 2 [9,10], membrane bound O-acetyltransferase domain containing 7 [11,12] have been associated with NAFLD. Environmental factors include sedentary life-style, a high-calorific and/or high-fat diet, especially rich in saturated fats [13], as well as environmental pollutants such as heavy metals, chlorination by-products, microcystins etc. [14]. In addition, various hepatotoxic-drugs such as valproic acid, aspirin, amiodarone and ibuprofen are also known to cause NAFLD by impairing lipid metabolism [15].

Despite its high prevalence, current treatments for NAFLD are limited to diet-alteration and lifestyle modification, which seems simple conceptually but very difficult to adhere to. Other pharmacological interventions currently being employed are treatment with insulin sensitizers such as Metformin and Thiazolidinediones. Insulin resistance exacerbates the pathogenesis of NAFLD and is commonly observed in these patients [16]. Metformin increases fatty acid β-oxidation and inhibits de novo synthesis of fatty acids [17,18]. Thiazolidinediones activate peroxisomal proliferator activated receptor γ (PPAR γ) and reduces hepatic fat content [19]. On the other hand, statins are prescribed as lipid lowering drugs [20,21,22]. Antioxidants as well as anti-inflammatory drugs are also commonly prescribed for NAFLD patients [3,23]. However, these strategies focus on alleviating the effect of NAFLD rather that resolving the pathogenesis of the disease. However, the fundamental mechanisms for pathogenesis of NAFLD are yet to be comprehended completely [24]. Thus, a cure for NAFLD remains elusive. Hence, it is important to address the underlying pathways involved in NAFLD progression and to identify molecular targets for better therapeutic approaches.

The initiating step of NAFLD is liver steatosis, the accumulation of excess lipids. By definition, steatosis ensues when more than 5% of normal liver weight is occupied by fats [25]. In a normal liver, lipid acquisition and resolution are tightly regulated. Any flaw in this regulation leads to an increase in hepatic lipid buildup i.e., steatosis. Increased fat uptake occurs due to fasting, a high-fat diet, or lipodystrophies. Similarly, increased blood glucose levels induce lipogenic genes in liver resulting in de-novo lipogenesis in liver. Increased inflow of fats is converted into triglycerides to reduce lipotoxicity of free fatty acids. However, in the case of overwhelming fat uptake, this hepatoprotective mechanism can be disrupted and leave hepatocytes steatotic and injured. An enhanced pool of free fatty acids may lead to oxidative stress, mitochondrial dysfunction, etc., adding to the dismay of injured hepatocytes [13]. These injured hepatocytes induce macrophages and Kupffer cells in liver to release pro-inflammatory as well as profibrotic signals leading to inflammation and development of fibrosis [13]. Fibrosis is the formation of scar tissue, which is a healing response to liver injury, wherein specialized liver cells known as hepatic stellate cells (HSCs) get activated and secrete extracellular matrix. However, deposition of extracellular matrix may lead to hardening of liver tissue and interference to normal liver function.

Intuitively, regulating lipid metabolism to avoid further repercussions of lipid accumulation such as NASH, liver injury and fibrosis seems to be a suitable strategy in managing and/or reversing NAFLD. To do so, it is important to identify underlying pathways and upstream regulators to manipulate these processes successfully. Thus, receptor tyrosine kinases (RTKs) with a robust track record as suitable molecular targeted therapeutics is worthy of closer examination. In this review, we will discuss some of the recent advances, projecting RTKs as a suitable target in the treatment of NAFLD. We will confine our review to early and potentially reversible stages of NAFLD such as simple steatosis, NASH and liver fibrosis [26,27,28,29].

## 2. Receptor Tyrosine Kinases (RTKs) and Its Role in NAFLD

The dysregulation of lipid metabolism provides an intrinsic opportunity to address the problem of steatosis, NASH, and even hepatic fibrosis. On this basis, RTKs are considered favorable for their role in regulating lipid metabolism and development of NAFLD. Broadly speaking, RTKs play a significant function in intra-cellular signaling and cell-to-cell communication. They respond to external signals such as growth factors and initiate a cascade of intracellular responses that regulate cell growth, proliferation, differentiation, motility, and other cell-type specific functions [30,31]. Individual cells and tissues in the organism exhibit tight regulation of RTKs via expression and phosphorylation-mediated controls. Aberrations in this signaling are associated with many cancers, neurodegenerative diseases, cardiovascular diseases and so forth. Being at the upper echelon of cell signaling where RTK binds directly to extracellular signals, they sit at the proximal end of many biological processes and hence at the center of diverse maladies.

NAFLD is no substitute for this fact, as various RTKs have been found to participate in liver development, liver regeneration as well as maintenance of liver function. For instance, epidermal growth factor receptor (EGFR) when activated by its ligand epidermal growth factor (EGF) is found to upregulate lipogenesis [32,33], while the hepatocyte growth factor (HGF)/mesenchymal-epithelial transition factor (cMET) pathway is found to check increased lipogenesis as well as improve β-oxidation of fatty acids [34]. The fibroblast growth factor 21 (FGF21)/fibroblast growth factor receptor 1 (FGFR1) axis is found to limit hepatic lipid accumulation by remote regulation from adipocytes [35]. On the other hand, another isoform of FGF receptor family, fibroblast growth factor receptor 4 (FGFR4), is found to exacerbate lipid accumulation by reducing β-oxidation of fatty acids [36]. In this review, we will address the metabolic functions of RTKs and their ligands, as well as the molecular mechanisms by which these RTKs regulate the development and progression of NAFLD. This knowledge will help us appraise and validate specific RTKs as drug targets for disease management

### 2.1. Epithelial Growth Factor Receptor (EGFR)

EGFR is highly expressed on hepatocytes [37,38]. Upon binding to ligands such as epidermal growth factor, EGFR regulates biological processes such as early liver development, liver regeneration, and repair [39,40]. Through various downstream pathways, EGFR is known to regulate lipid metabolism in the liver [32,41] and its dysregulation is found to be associated with lipid accumulation and NAFLD. The effect EGFR on lipid accumulation could be an associated consequence of its role in liver regeneration. During liver regeneration, a transient lipid accumulation is observed which is abolished with EGFR inhibition. Yet, EGFR inhibition shows a minor effect on hepatocyte proliferation. Hence, the impact of EGFR perturbation may be weighted more towards lipid metabolism than proliferation during liver regeneration [42].

In a recent study by Bhushan B. et al. (2019), a microarray analysis has shown that EGFR inhibition altered approximately 40% of genes which are found to be dysregulated under a fast food diet. This finding established the molecular underpinnings for the role of EGFR in lipid accumulation in hepatocytes [33] Biochemically, EGFR controls enzymes involved in lipid metabolism by regulating transcriptional factors such as Sterol regulatory element binding protein1c (SREBP1c), Carbohydrate-response element binding protein (ChREBP), peroxisome proliferator-activated receptor γ (PPARγ) and hepatocyte nuclear factor 4-α (HNF4α) [40]. SREBP1c is a major transcriptional factor controlling expression of genes involved in lipogenesis. Upstream of this, EGFR controls expression of SREBP1c via Phosphoinositide 3-kinases (PI3K)/protein kinase B (AKT)/mammalian target of rapamycin (mTOR) pathway [32,43,44,45,46]. Moreover, high fat diet results in overexpression of PPAR-γ, which induces SREBP1c and its downstream pathway leading to lipogenesis [33,47]. Correspondingly, inhibition of EGFR results in an increased expression of HNF4α which checks PPAR-γ and associated lipogenesis [33]. EGFR inhibition is also associated with improved hyperlipidemia. It controls proteins such as microsomal triglyceride transfer protein (MTP) and apolipoprotein B (ApoB) which are involved in VLDL secretion. Increased VLDL secretion results in hyperlipidemia. However, blocking of EGFR is found to be a suitable strategy to control hyperlipidemia [48]. Overall, these reports support the therapeutic benefit of EGFR inhibition towards reducing lipid accumulation in the liver. Target validation using RTK inhibitors such as AG1478, Gefitinib, PD153035 further corroborated their impact on reducing lipogenesis and thus steatosis [32,49].

Mechanistically, targeting EGFR signaling appears to be a promising strategy in lowering hepatic oxidative stress and the damage that follows. Liang D. et al. (2018) have shown that mice fed on high fat diet had increased levels of NOX-dependent superoxide production. In their study, EGFR inhibitors such as AG1478 and compound 451 blocked the production of these superoxides and reduced oxidative stress [49]. In another study, EGFR inhibitor PD153035 was found to reduce inflammatory markers [32]. Additionally, EGFR has strongly been associated with liver fibrosis. Its activation due to high fat diet causes HSCs to secrete fibrotic markers such as α-smooth muscle actin (α-SMA) and collagen, promoting fibrosis [33,49]. Importantly, this effect reverses upon EGFR inhibition. The significance of this finding is that it ascertains the role of EGFR in the development of fibrosis as a sequelae of a high fat diet. The effect of EGFR on HSC activation could be linked to crosstalk between transforming growth factor-β (TGF-β)/SMAD pathway and EGFR/extracellular-signal-regulated kinase (ERK) pathway [50,51]. Activation of EGFR pathway increases expression of TGF-β, a central regulator of fibrogenesis responsible for HSC activation and epithelial-mesenchymal transition (EMT) [51].

All of these reports claim that EGFR is involved in NAFLD development and progression through activation of lipogenic genes, development of oxidative stress and inflammatory response, as well as activation of HSCs. Inhibition of EGFR leading to prevention of fibrosis in high fat diets demonstrates how EGFR could be a nexus that distinguishes asymptomatic steatosis from one that progresses into liver fibrosis. This correlation reinforces the potential of EGFR in the treatment of NAFLD. Various well characterized EGFR inhibitors have already been studied and approved for their use in cancer treatment. However, to translate the use of EGFR inhibitors in NAFLD clinically, we must evaluate the non-targeted effects of EGFR inhibition on other tissues.

### 2.2. Hepatocyte Growth Factor Receptor

HGF plays a pivotal role in liver regeneration. It acts by binding to an RTK named c-MET. HGF/c-MET signalling pathway regulates vital cellular processes such as cell growth, migration, mesenchymal to epithelial transition etc. [52]. The HGF/cMET pathway has also been associated with NAFLD. HGF is reported to stimulate lipogenesis as well as lipid mobilization [53,54]. As HGF was found to induce lipid synthesis via c-MET, blocking HGF/c-MET pathway by inhibition of c-MET could be a plausible strategy. Yet, paradoxically, recent reports suggest that activation of c-MET pathway by HGF supplementation alleviates steatosis [55,56]. This contradiction in results could be attributed to the dual ability of HGF to induce lipid synthesis and to resolve accumulated lipids by activation of lipid oxidation. HGF regulates farnesoid-X receptor (FXR) pathways and controls peroxisome proliferator activated receptor-α (PPAR-α), which facilitates fatty acid oxidation. [56,57,58]. The effect of HGF on β-oxidation was corroborated by Kroy D. et al. (2014) using cMET knockout mice fed on a Methionine/Choline Deficient (MCD) diet. They have found that cMET deletion was associated with reduced β-oxidation of fatty acids which in turn lead to the accumulation of fatty acids [34]. HGF treatment also induces genes involved in lipid secretion such as MTP and ApoB [59]. However, this mechanism is not thoroughly addressed in diet induced fatty liver except one study conducted by Kosone and team. In 2007, Kosone T. et al. demonstrated that HGF supplementation increased lipid secretion, and it was achieved by augmented expression of the ApoB via MAPK pathway [55]. Moreover, HGF also reduces the expression of fatty acid uptake proteins such as CD36 [34]. Based on this, the HGF/c-MET pathway gains its significance in NAFLD treatment as it controls liver lipid metabolism on various fronts starting from fatty acid uptake to the resolution of excess fats by controlling β-oxidation and VLDL secretion.

The HGF/cMET axis is also important in controlling the progression of NAFLD to NASH. HGF shows an anti-inflammatory effect during a high fat diet by mitigating the expression of inflammatory cytokines such as TGF-α, Interlukin-6(IL-6), Interleukin 1 β (IL-1β) etc. [57,60]. Marqyardt J. et al. (2012) have done an elaborate work to explain the role of c-MET in attenuating NASH by using c-MET knockout mice model. In their study, c-MET deletion reduced the expression of nuclear factor-erythroid factor 2-related factor 2 (Nrf-2), while it enhanced the expression of nicotinamide adenine dinucleotide phosphate (NADPH) oxidase complex and increased production of the reactive oxygen species. It thus increased oxidative stress. Their transcriptome analysis showed that c-MET deletion reduced the expression of genes involved in the stress response, DNA damage, oxidative stress regulation and hence impairs the regenerative capacity of the liver [61]. In addition, c-MET deletion also led to activation of HSCs and the development of fibrosis [61,62].

In short, the deletion of cMET and hampering of the HGF/cMET axis leads to the development of steatosis. It may also exacerbate NASH or uncontrolled fibrosis. Thus, unlike EGFR, activation, rather than the inhibition of the HGF/cMET pathway, will be the strategy to manage NAFLD. However, as HGF increases secretion of lipids, it is crucial to monitor the adverse side effects of hyperlipidaemia. Besides, HGF is also associated with increased risk of tumour formation and metastasis in HCC [63]. As a result, before considering its therapeutic application, the use of HGF supplementations must be justified with more experimental evidence, and the challenge of formulating protein-based drugs must be appropriately investigated.

### 2.3. TAM (Tyro3, AXL, MERTK) Receptor

The TAM receptor tyrosine kinase family is made up of three distinct receptors, namely Tyro3, AXL and MERTK. These receptors are activated by the Growth arrest-specific gene 6 (GAS6) and protein S. Gas6 is mainly associated with hepatic metabolism through TAM receptors. However, these TAM receptors exhibit differential expressions and functions. In liver, AXL are expressed in macrophages, HSCs and endothelial cells [64]. AXL expression on hepatocytes was upregulated in malignant hepatocytes [65]. MERTK is present on Kupffer cells and sinusoidal endothelial cells [66]. Whereas Tyro3 is expressed mainly on resident macrophages, while its overexpression has been reported in hepatocytes during HCC [66,67]. Amongst all three receptors, AXL exhibits higher affinity towards Gas6 and the Gas6/AXL pathway is known to be involved in the development of steatosis to fibrosis.

The increased activation of AXL is observed during high fat diet-induced steatosis, suggesting its correlation with the development of NAFLD. Fourcot A. et al. (2011) carried out a study to understand involvement of GAS6 and its receptors in NAFLD. They observed that blocking of the Gas6/AXL pathway significantly reduced triglyceride content, and overall steatosis score in mice fed on a high fat diet. Further, they have demonstrated that this improvement in steatosis involves an increased expression of transcriptional factor PPAR-α and two rate limiting enzymes, namely, acyl-CoA oxidase-1 (ACOX1) and carnitine palmitoyltranferase-1 (CPT1), which are critical to mitochondrial β-oxidation [64]. In a recent study conducted by Tutusaus A. et al. (2020), AXL inhibition using Bemcentinib was found to reduce lipid accumulation and overall NAFLD score, in mice fed on high fat diet [68]. Furthermore, AXL inhibition was also found to reduce the expression of proinflammatory cytokines, tumour necrosis factor (TNF) or monocyte chemoattractant protein-1 (MCP-1) [64,68].

AXL prominent expression in HSCs alludes to its profibrogenic effect. The Gas6/AXL pathway activates the PI3K/AKT pathway. It further induces NF-kB p65 translocation to the nucleus for antiapoptotic response against HSCs, thereby improving survival, proliferation and the activation of HSCs and promoting fibrosis. This sequalae was found to reverse by the AXL inhibitor, bemcentinib [69,70]. In a study conducted by Bárcena C. et al. (2015), the authors used both a genetic model of Axl deficiency (Axl KO) and a chemical inhibition of AXL by BGB324 and observed reduced HSC activation, thus, proving the role of Gas6/AXL pathway in fibrosis development [70].

AXL, upon activation, undergoes extracellular domain cleavage by A Disintegrin and Metalloproteinases (ADAM)10 and ADAM17, yielding soluble version of the proteins sAXL. sAXL is present in serum and its expression is found to increase in liver diseases [71]. While this feature is also observed with other TAM receptors, only sAXL expression was found to increase during steatosis [65,69]. This confirms involvement of AXL in steatosis. Independently, sAXL also has an ability to bind to Gas6 and thus depleting availability of Gas6 to bind to AXL. This attenuates the progress of Gas6/AXL pathway which leads to fibrogenesis. However, Gas6 level is augmented in liver diseases. Holstein E. et al. (2018) explained this disparity in data by claiming that inhibitory effect of sAXL is abolished due to abundance of non-shredded AXL, which can still continue to bind to Gas6 [72]. Tutusaus and team (2020) suggested that AXL inhibition in fact induces Gas6 upregulation as compensatory mechanism. Further, Gas6 may exhibit a hepatoprotective effect against the development of liver ailments [68]. Thus, inhibition of AXL not only blocks the progression of NAFLD but also triggers the hepatoprotective role of Gas6.

The hepatoprotective effect of Gas6 was observed by Llacuna L. et al. (2010). They have shown that Gas6 plays hepatoprotective role in ischemia and hypoxia induced liver model [73]. Tutusaus’s study also vouches for hepatoprotective role of Gas6. In their study, Gas6/MERTK protected hepatocytes from palmitic acid induced lipotoxicity by activation of AKT/signal transducer and activator of transcription 3 (STAT3) pathway [68]. On the contrary, a study conducted by Cavali M. et al. (2017) and Cai B. et al. (2020) demonstrated that MERTK inhibition was associated with reduced fibrosis [74,75]. Thus, further studies are required to warrant the exact role of MERK as hepatoprotective or as profibrogenic. Nevertheless, it is vital to recognize the potential for AXL and MERTK to play differential roles in NAFLD, necessitating the use of specialized inhibitors. Otherwise, co-inhibition of AXL and MERTK, on the other hand, could result in opposite actions with unfavorable effects. The third TAM receptor, Tyro3, is highly upregulated in HCC patients [67]. However, to the best of our knowledge there are no reports available discussing its role in the development and progression of steatosis.

In summary, TAM receptors present novel therapeutic potential in steering the progression of steatosis towards NASH and fibrosis by manipulating inflammatory and fibrotic factors. Their expression can be considered as biomarkers to identify NAFLD with a propensity for disease progression. That said, its exact role in lipid metabolism and contribution to steatosis remains to be ascertained.

### 2.4. Fibroblast Growth Factor Receptor

Fibroblast growth factor receptor (FGFR) is a family of four RTKs (FGFR1–4). As the name suggests, FGFRs bind to fibroblast growth factors (FGFs) (1–22) to facilitate a plethora of cellular signaling pathways controlling cell growth, differentiation, and metabolic activities. In recent years, FGFRs were found to be involved in liver development, homeostasis, as well as regeneration. Moreover, FGFR aberrations have also been correlated with liver diseases such as steatosis, fibrosis and even carcinogenesis.

Different FGFR subtypes exhibit a non-redundant role within the liver. FGFR1 is predominantly expressed on adipocytes and it regulates adipocyte-hepatocyte communication to maintain hepatic lipid metabolism. FGF19 and FGF21 are two potent ligands for FGFR1. FGF21 is a stress hormone, which is produced and upregulated in liver during hepatic stress. FGF21 binds to FGFR1 on adipocytes and modulates lipolysis in adipocytes and lipogenesis in hepatocytes to maintain lipid homeostasis. Earlier, Yang C et al. (2012) have shown that adipocyte specific deletion of FGFR1 mediates indirect effect on hepatic lipogenic genes. Under hepatic stress, adipocyte ablation of FGFR1 leads to an increase in hepatic lipogenic genes such as SREBP 1c, diacylglycerol O-acyltransferase 1 (DGAT1), acetyl-coA carboxylase (ACC), fatty acid synthase (FAS), stearoyl-coenzyme A desaturase (SCD1) and PPARγ [76]. In another study, Xu J. et al. (2009) have shown that activation of FGFR1 by FGF21 supplementation can reduce expression of SREBP 1c, ACC, FAS, PPARγ in mice fed on HFD [35]. The role FGFR1 in maintaining healthy liver condition and avoiding development of steatosis was indirectly advocated by Hu Y. et al. (2020). In their work, it was demonstrated that miR-22 blocks FGF21 and FGFR1 signaling. However, inhibiting miR-22 resulted in restored FGFR1 signaling and its control over lipid metabolism in liver. This resulted in the attenuation of steatosis in mice kept on a high fat diet. This suggests that selective activation of the FGFR1-mediated pathway in early stages of NAFLD can have therapeutic promise [77]. However, activation of FGFR1 must be thoroughly studied for potential side effects. While FGFR1 is a boon against liver steatosis, it acts as a bane in later stages of liver diseases, such as liver fibrosis. It is reported that FGFR1 is more abundantly expressed on quiescent HSCs than on any other type of liver cells, and plays an important role in the development of fibrosis. Recent pieces of evidence suggest that FGFR1 induces nuclear factor-κB (NF-κB) and thus leads to increased release of inflammatory cytokines, proliferation of HSCs and development oof fibrosis. Reciprocally, the physiological inhibition of FGFR1 was found to attenuate liver fibrosis [78,79,80]. Earlier, we have seen that activation of FGFR1 on adipocytes controls the lipid metabolism in liver, whereas the, inhibition of FGFR1 on HSCs dampens fibrotic markers. Because of the differential role of FGFR1 in various stages of NAFLD, it is critical to evaluate organs and cell types that express the RTK to improve the viability of RTK inhibition therapy in managing the disease.

Another well-studied isoform of FGFRs of relevance to the liver is FGFR4, which is found abundantly on mature hepatocytes. FGFR4 is activated by its ligand FGF19 and induces triglyceride accumulation in a high fat diet [81]. In an earlier study conducted by Huang X. et al. (2007), during a high fat diet, FGFR4-/-phenotype altered lipid metabolism and reduced steatosis as compared to wild type [82,83]. FGFR4 deficiency is associated with elevation in expression of genes involved in lipid catabolism such as PPAR-α and lipid secretion such as MTTP and ApoB [82]. In addition to effect of FGFR4 inhibition on lipid catabolism and secretion, it also downregulated de-novo lipogenesis by reducing expression of genes involved in triglyceride synthesis such as SREBP1c, ACC, FAS, DAGT1 [82].

The involvement of FGFR4 in steatosis arises from its activation by FGF19. The FGF19 ligand has a dual role in lipid metabolism. On one hand, FGF19 when bound to FGFR4 was found to increase triglyceride accumulation. On the other hand, FGF19 displays a lipid lowering effect via binding to other FGFRs [81]. In a study conducted by Yu X. et al. (2013), it was observed that FGFR4 inhibition leads to the increased expression of FGF15, an ortholog of FGF19 in rodents. Further, they have proved that FGF19 supplementation comparable to the level of FGF15 produced in FGFR4 deficient cells a reduced lipid load in diet-induced obese mice [84]. These results question the established understanding that FGF19 activates FGFR4. However, it is also understood that once activated, FGFR4 controls hepatic expression of Cholesterol 7-α hydroxylase (CYP7A1), and thus regulates bile synthesis and reduces FGF19 expression. This could justify the results where FGFR4 inhibition increased expression of FGF15/19. Wu X. and team (2013) have demonstrated that FGFR4 is essential for FGF19 to suppress CYP 7a, but FGFR4 is not required by FGF19 in lipid metabolism [81]. This study established a link between FGFR4 deficiency, increased expression of FGF15/19, and the lipid lowering effect of FGF15/19 in the absence of FGFR4. Thus, FGFR4 inhibition appears to be a promising strategy, but with an increased hazard of hyperlipidemia development. Hence, the use of FGFR4 inhibition needs to be warranted for its use in a steatosis treatment regime.

### 2.5. Vascular Endothelial Growth Factor Receptor (VEGFR)

VEGF is an important signaling protein involved in both vasculogenesis and angiogenesis. The effect of VEGF is facilitated by its binding to two RTKs, VEGF receptor-1 (Flt-1) and VEGF receptor-2 (Flk-1/KDR). This signaling is critical to hepatic angiogenesis, which in turn is closely associated with progression of liver fibrosis. Tarantino and team (2009) shed light on increased VEGF during NASH as a diagnostic marker [85]. In addition, a few more reports claim that VEGF is upregulated and involved in the progression of NAFLD from early stages such as steatosis [86,87,88].

Coulon S. et al. (2012) have reported clinical data showing increased levels of VEGFR in NAFLD and NASH patients as compared to controls [86]. Their study further provides experimental evidence that VEGF is increased during the transition from steatosis to NASH in a mice model fed on MCD diet, while inhibition of VEGF receptors (VEGFR2) arrested the development of NASH by downregulating genes involved in inflammatory response such as tumor necrosis factor-α (TNF-α), interleukin-1b (IL-1b). It also upregulated expression of scd1, a gene involved in converting excess lipids in monounsaturated fatty acids which can be stored safely without causing lipotoxicity. This contributed to protection against lipotoxicity in a high-fat diet and stopped onset of steatohepatitis [87]. At this moment, there are no mechanistic details available on how VEGFR affects lipid accumulation in the liver. Hence, this observed phenotype of VEGFR inhibition has to be investigated further for its effect on pathways maintaining lipid metabolism in the liver.

As discussed earlier, VEGF/VEGFR signaling plays a key role in the pathogenesis of liver fibrosis [89,90]. VEGF is secreted by hepatocytes during liver injury [90,91]. It acts through VEGFR-2 present on sinusoidal endothelial cells and HSCs to promote angiogenesis followed by fibrosis. VEGF, when bound to VEGFR-2, activates HSCs via the PI3K/AKT pathway and increases fibrotic markers such as α-SMA and collagen in a process of fibrinogenesis [90,92]. Thus, while its targeted role in anti-steatosis remains to be established, blocking VEGF/VEGFR-2 signaling holds great promise in controlling liver fibrosis.

## 3. Downstream Pathways through Which RTKs Regulate NAFLD

RTKs are known to be involved in the development and progression of NAFLD via various pathways such as PI3K/AKT/mTOR, RAS/ERK, Janus Kinase (JAK)/signal transducer and activator of transcription proteins (STAT3), FXR/SHP, etc. These pathways present as cascading signals from RTKs to effector molecules such as nuclear receptors, transcriptional factors or other vital intermediates controlling lipid metabolism, oxidative stress, inflammation and also fibrosis. Figure 1 summarizes the complex network of downstream pathways through which RTKs control various characteristics of NAFLD.

Among these, the PI3K/AKT pathway is found to be regulated by various RTK, such as EGFR, AXL, and VEGFR [58,60,69,70,90,92,93]. The activation of the PI3K/AKT pathway has already been reported in NAFLD patients. Phosphorylation of PI3K/AKT leads to activation of mTOR, which is known to regulate the transcriptional factor SREBP1c and control lipogenesis [93]. mTOR is also known to inhibit autophagy or, to be specific, lipophagy, which may also contribute to lipid accumulation [94,95]. Additionally, PI3K/AKT leads to the activation of transcriptional factor NF-κB, resulting in increased inflammation and fibrosis [96]. In short, the PI3K/AKT pathway is regulated by various RTKs and is identified as one of the prime pathways regulating various stages of NAFLD, starting from lipid accumulation to inflammation and fibrosis.

Among other pathways involved in NAFLD, RAS/ERK and JAK/STAT3 pathways are profoundly involved in the development of fibrosis [97,98,99,100]. As discussed earlier, RTKs such as EGFR, c-MET and FGFR1 activate the RAS/ERK pathway. Furthermore, the RAS/ERK pathway is known to regulate the activation and proliferation of HSCs via SMAD-dependent and SMAD-independent pathways [101,102], whereas, EGFR, MERTK and FGFR4 induce JAK/STAT pathways to upregulate fibrotic markers. Lastly, FXR/SHP signalling regulated by c-MET and FGFR4 is known to control the expression of transcriptional factor PPAR-α and thus monitor lipid catabolism by β-oxidation and VLDL secretion.

As depicted in Figure 1, RTKs control more than one cellular function via various downstream pathways. Similarly, each regulatory pathway is differentially regulated by different RTKs. While these signalling pathways diverge as well as converge within the cells, specific RTK targets may assert greater influence on NAFLD development and progression by affecting more than one downstream pathway. Amongst all the RTKs discussed, EGFR appears to regulate various stages in NAFLD, starting from lipid accumulation to inflammation, oxidative stress and even fibrosis. However, this does not end the hunt for a viable RTK target. That said, our knowledge of the regulatory effects of other RTKs is limited, and more pieces of evidence are needed to unravel the potential of other RTKs to regulate NAFLD at various stages. But the emerging picture does provide an opportunity to explore the synergistic effect of RTK inhibitors. This may mitigate the chances of drug failure due to mutations in drug targets, mutations in effector signalling, and bypass signalling, which imposes resistance against drugs.

## 4. Strategies to Target RTKs

In this survey of RTKs as regulatory controls of NAFLD, we arrive at a picture where activation of some RTKs (HGF, FGFR1) and inhibition of others (EGFR, AXL, FGFR4 and VEGFR) have found to be useful in tackling NAFLD. But as a therapeutic strategy, activating RTKs via the introduction of suitable growth factors may impose more challenges, as its efficacy is subjected to the inherent expression of RTKs on targeted cells. Besides, overexpressed and highly activated forms of RTKs and their growth factors are associated with various malignancies and present unforeseen risks, especially with prolonged treatment. Hence, the inhibition of RTKs provide a suitable therapeutic option. Various RTK inhibitors have proven to be a great success in oncogenic as well as non-oncogenic therapies. These inhibitors not only provide treatment options but also shed light on the cellular microenvironment and signalling pathways for deeper understanding. Some of the commonly discussed RTK inhibitors can be grouped as small molecule inhibitors, therapeutic antibodies, natural products, and nanoparticles. However, all these categories of RTK inhibitors come with their own advantages and disadvantages.

### 4.1. Small Molecule Inhibitors

As of March 2021, 62 small molecule RTK inhibitors have been clinically approved by the FDA [103]. EGFR and VEGFR are one of the common targets for these small molecule inhibitors. Many of them are deployed in cancer treatments [103]. However, a few experimental reports mentioned in Table 1 demonstrate the use of RTK inhibitors such as Gefitinib, Bemcentinib and Sunitinib targeting EGFR, AXL and VEGFR respectively in NAFLD management. Although these experimental results explain the effectiveness of RTK inhibition by small molecules, only two have reached clinical stage for example Erlotinib, an EGFR inhibitor and Sorafenib, a VEGFR inhibitor for improving liver fibrosis. Thus, it is important to take a step further and study small molecule inhibitors for clinical use in NAFLD treatment.

### 4.2. Therapeutic Antibodies

Therapeutic antibody treatment is another useful strategy to consider in NAFLD management. Therapeutic antibodies provide high specificity towards the target to avoid off-target side-effects. Recent antibody engineering advances could design antibodies with wider target ranges and with more efficient and long-lasting effects. Therapeutic antibodies could be modified and refined for industrial production as well. Thus, it is worth considering therapeutic antibodies as a tool in treatment of NAFLD. Various monoclonal antibodies have been discussed to selectively target VEGFR, EGFR etc. in cancer therapies [109]. However, RTK targeting antibodies are seldom studied for their use in NAFLD management. As explained in Table 1, anti VEGFR-2 antibodies (DC101) are one such rare example. Coulon S. et al. (2013) have found that high fat diet-induced steatosis and NASH were successfully resolved upon treatment with anti-VEGFR-2 antibodies (DC101) [87]. Furthermore, an in depth study has to be done to identify such RTK targeting antibodies to develop selective and effective treatments against NAFLD.

### 4.3. Natural Products

Natural products provide a pool of compounds to be considered as drugs in NAFLD management. Due to their lower toxicity, better chemical diversity, and inherent link to biological targets, natural products obtain the attention of the scientific community in various treatment regimens [110,111]. Various natural products have been found to be useful in alleviating NAFLD, from simple steatosis to liver cirrhosis. Curcumin [112,113], Resveratrol [113,114], Luteolin [115,116], and Honokiol [117] are some of the instances of natural products which have been studied for their anti-steatotic and anti-fibrotic effects. While the exact mechanism of action remains elusive for many of these compounds, RTK inhibition has been established for some of them. Curcumin has been identified to target EGFR, AXL, and FGFR, making it a multi-targeting RTK inhibitor [118,119,120]. A phenolic compound, Honokiol is known to inhibit EGFR signalling. It is also found to augment inhibitory effect of other EGFR inhibitors such as Erlotinib and Lapatinib in cancer treatment [121,122]. However, there is a scientific gap where hardly any natural product has been studied to establish the relation between RTKs and NAFLD progression. This provides an opportunity to identify drug compounds from natural products to target RTKs in NAFLD treatment. However, natural products come with their own limitations, such as difficulty in isolating active compounds and low solubility, etc.

### 4.4. Nanoparticles

Nanoparticles have recently grabbed wide attention as a vehicle to improve the target-based delivery of various drugs. Nevertheless, some nanoparticles also bear various pharmacological properties which makes them a good choice of drugs in NAFLD treatment. Various inorganic nanoparticles (NPs) have shown to be protective against NAFLD. Cerium oxide and zinc oxide NPs have been shown to reduce lipid accumulation and thus control liver steatosis [123], while, titanium dioxide NPs, silicon dioxide NPs, manganese NPs and gold NPs have displayed anti-fibrotic effects [124,125]. On the other hand, nanoparticles also possess the ability to block RTK signalling pathways. Titanium dioxide NPs, silicon NPs, and gold NPs are known to target VEGFR [126,127], while FGF1-loaded gold NPs are known to target FGFR in cancer therapy [128]. Thus, it is logical to consider NPs for their santi-RTK effects to be used in NAFLD treatment.

In summary, small molecule inhibitors, therapeutic antibodies, natural products, and nanoparticles are viable options for treating NAFLD. Currently, various drugs are clinically approved for their anti-RTK effects. Not to mention, various compounds are continuously being reported to be effective against NAFLD. Thus, it is time to connect the dots and identify RTK targets and suitable drugs targeting RTK for better management of NAFLD. Moreover, the synergistic effects of RTK inhibitors boost the likelihood of viable NAFLD treatment.

## 5. Conclusions

In conclusion, the inhibition of RTKs holds great potential as a therapeutic approach in NAFLD treatment. However, we must remain cognizant that most of the available data projecting RTKs as suitable targets in NAFLD is based on animal research. Numerous animal models such as diet-induced (high fat diet, methionine-deficiency diet, choline deficiency diet), chemical-induced (Streptozotocin, Carbon tetrachloride) as well as genetic (leptin deficiency model) are being used in NAFLD research. However, each of these models presents unique challenges that limits the recapitulation of human physiology [129]. Also, animal studies may not precisely predict the toxic effects of drugs in humans [130]. We must not overlook the long-term side effects of drugs. Liver is the principal organ to metabolize drugs. During NAFLD, a few metabolic pathways may undergo alterations, perturbing drug disposition and causing toxicity. On the other hand, defective drug metabolism can exacerbate NAFLD progression [131]. Thus, it is imperative to understand the complex relationship between NAFLD pathogenesis and drug metabolism, while considering a drug candidate. All of these issues pose challenges to the prospect of RTKs in NAFLD treatment. In short, without oversighting the suitability of RTKs in NAFLD management, we must critically analyze the available data and interpret them with caution.

In summary, this review provides a comprehensive view of RTKs that have been explored for NAFLD management, and rationalizes their application by appraising the underlying pathways that they control. Specifically, the inhibition of EGFR, AXL, FGFR4 and VEGFR were found to be viable treatment options based on mechanistic evidence across various in vitro and in vivo studies. EGFR is found to regulate NAFLD at various stages, starting from steatosis to NASH, and even with regard to fibrosis, while other RTKs discussed, such as AXL, FGFR4 and VEGFR, are also found to be effective in regulating different manifestations of NAFLD. Investigation of these targets can also leverage the availability of potent and selective FDA-approved inhibitors. Overall, this review accentuates the role of RTKs in the NAFLD management regime. Thus, in the fight against NAFLD, it is compelling to explore some of these RTKs for in-depth investigations.

## Figures and Tables

**Figure 1 biomedicines-09-01776-f001:**
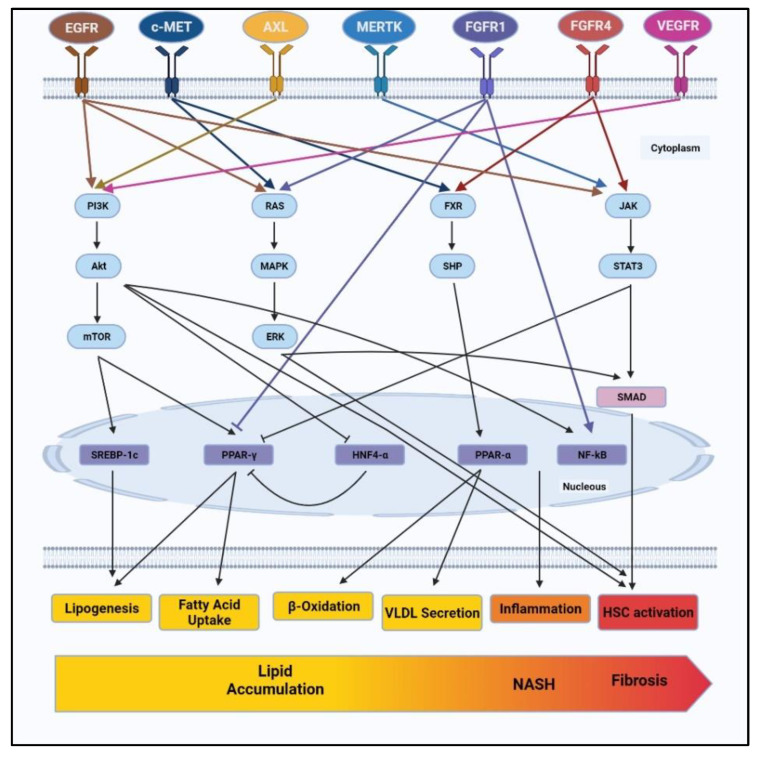
RTKs such as EGFR, c-MET, AXL, MERTK, FGFR1, FGFR4 and VEGFR control signaling pathways involved in NAFLD progression. RTKs regulate vital cellular pathways such as PI3K/AKT, RAS/ERK, FXR/SHP and JAK/STAT3. Navigating through these pathways, RTKs regulate lipid metabolism including lipogenesis, fatty acid uptake, β-oxidation, VLDL secretion. RTKs further involved in development of liver inflammation and HSCs activation to cause NASH and fibrosis. NAFLD: non-alcoholic fatty liver disease, EGFR: epidermal growth factor receptor, c-MET: mesenchymal–epithelial transition factor, MERTK: MER proto-oncogene tyrosine kinase, FGFR1: fibroblast growth factor receptor 1, FGFR4: fibroblast growth factor receptor 4 and VEGFR: vascular endothelial growth factor receptor, PI3K: phosphoinositide 3-kinases, AKT: protein kinase B, ERK: extracellular-signal-regulated kinase, FXR: farnesoid X receptor, JAK: janus kinase, STAT3: signal transducer and activator of transcription 3. VLDL: very low density lipoprotein, HSCs: hepatic stellate cells.

**Table 1 biomedicines-09-01776-t001:** RTK Inhibitors showing improved steatosis and/or fibrosis in experimental set-ups. RTKs: receptor tyrosine kinases, EGFR: epidermal growth factor receptor, FGFR4: fibroblast growth factor receptor 4 and VEGFR: vascular endothelial growth factor receptor.

RTK	Inhibitor	Study	Reference
EGFR	Gefitinib	Gefitinib attenuated palmitic acid induced lipid accumulation in Huh7 cells by inhibiting lipogenic genes	[32]
	Erlotinib	Erlotinib attenuated HSC activation and fibrosis after liver injury in mice treated with C 1Cl4 and bile duct ligation.	[40,104]
	AG1478	AG1478 reduced diet-induced fat accumulation as well as HSC activation and proliferation	[49,105]
	PD153035	PD153035 controlled lipid accumulation in high fat fed mice by downregulating lipogenic genes	[32]
AXL	Bemcentinib (BGB324)	Bemcentinib inhibits AXL and reduces liver inflammation and fibrosis in diet induced mouse model by inactivation of AXL/AKT phosphorylation and blocking of successive HSC activation	[68,70]
FGFR4	Soluble FGFR4 extracellular domain fragment	Blocking FGFR4 by soluble extracellular domain leads to decrease in steatosis.	[83]
VEGFR	DC101	Treatment with anti VEGFR-2 antibodies (DC101) reduced steatosis, inflammation as well as fibrosis in mice fed on MCD diet.	[87]
	PTK787/ZK222584 (PTK/ZK)	Inhibits HSC activation by attenuating HSC proliferation, migration, and collagen synthesis through the VEGF pathway	[92,106]
	Sunitinib	Treatment with Sunitinib resulted in decrease in inflammatory infiltrates as well as fibrotic markers such as α-SMA and collagen through VEGF pathway	[107,108]

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
