# Peer review of "Exploring the Gamut of Receptor Tyrosine Kinases for Their Promise in the Management of Non-Alcoholic Fatty Liver Disease"

_biomedicines, 2021, doi:10.3390/biomedicines9121776_

Round 1
Reviewer 1 Report
In the manuscript " Exploring the gamut of receptor tyrosine kinases for their promise in the management of non-alcoholic fatty liver disease", the authors Sayali Bhave and Han Kiat review the literature on the role of receptor tyrosine kinases in the development of NAFLD and the ways they can be targeted to manage the disease. The review is well written and there are enough figures to summarize the points discussed by the authors. The authors have discussed both the advantages and drawbacks of using inhibitors or activators of receptor tyrosine kinases to manage NFLAD. The review is sufficiently interpretive and the authors have posted some important questions for future research. There are a few typos that need to corrected. Also, in line 67 in the introduction “treatment with Metformin and Thiazolidinediones which blocks peroxisome proliferator-activated receptor γ (PPARγ) and increases insulin resistance” I think the authors mean improves insulin sensitivity since the sentence does not convey the correct information. The manuscript can be accepted pending minor corrections.
Author Response
We thank the reviewer for their valuable comments and suggestions. We have incorporated suggested changes in the manuscript. Here is a point-by-point response to the comments.
Comment 1: There are a few typos that need to be corrected.
Response: Thank you for bringing our attention to typing errors. We will wet through our review once again and avoid any typing mistakes.
Comment 2 : Also, in line 67 in the introduction “treatment with Metformin and Thiazolidinediones which blocks peroxisome proliferator-activated receptor γ (PPARγ) and increases insulin resistance” I think the authors mean improves insulin sensitivity since the sentence does not convey the correct information.
Response: Indeed, Metformin and Thiazolidinediones improve insulin sensitivity. Thank you for pointing out the error in the write-up. We have made suitable changes in the write-up from lines 68-73.
We once again thank the reviewer and editor for giving us an opportunity to make changes to our manuscript.
Reviewer 2 Report
Authors should be congratulated for approaching an interesting topic that is the NAFLD therapy.
Comments&Suggestions that authors should take into account.
Authors state that ....Metformin and Thiazolidinediones increase insulin resistance......in lines 66-67.
Indeed, metformin increases insulin sensitivity, enhances peripheral glucose uptake, increases fatty acid oxidation, and decreases absorption of glucose from the gastrointestinal tract, as evident in ....Metformin increases insulin sensitivity and basal glucose clearance in type 2 (non-insulin dependent) diabetes mellitus. Aust N Z J Med. 1991 Oct;21(5):714-9.
Lean NAFLD should be more precisely defined evaluating the abdominal adiposity, datum not always present when dealing with this type of NAFLD, as evident in....Lean-non-alcoholic fatty liver disease increases risk for metabolic disorders in a normal weight Chinese population. World J Gastroenterol. 2014 Dec 21;20(47):17932-40.
Authors correctly state that ....However, these strategies focus on alleviating the effect of NAFLD rather that resolving the pathogenesis of the disease...but they should quote the pertinent reference concerning the fact that the inner mechanisms of NAFLD are far from being clarified that is....J. Clin. Med. 2020, 9(1), 15; https://doi.org/10.3390/jcm9010015.
Authors state that ...Moreover, literature also reports upregulation of VEGF during early stages of NAFLD such as steatosis....
But, one of the first reports on the role of this main cytokine in human NAFLD was .....Could inflammatory markers help diagnose nonalcoholic steatohepatitis? Eur J Gastroenterol Hepatol. 2009 May;21(5):504-11.
Authors should state that the most part of studies that they present are performed using animal models of NAFLD, but they do not completely mirror human NAFLD as evident in.... Animal Models of Nonalcoholic Fatty Liver Disease-A Starter's Guide. Nutrients. 2017 Sep 27;9(10):1072. doi: 10.3390/nu9101072. PMID: 28953222; PMCID: PMC5691689.
Thus, plenty of findings should be taken with a pinch of salt and a note of caution is needed.
The long-term effects concerning the use of these new molecules should carefully followed and ascertained, as well as the possible drug-drug interactions, as evident in...Molecular Interactions between NAFLD and Xenobiotic Metabolism. Front Genet. 2013 Jan 22;4:2.
These last points should be at large discussed to offer readers a broader view of the topic.
Author Response
We thank you for your valuable comments and suggestions. We have incorporated suggested changes in the manuscript. Here is a point-by-point response to the comments.
Comment1: Authors state that ....Metformin and Thiazolidinediones increase insulin resistance......in lines 66-67.
Indeed, metformin increases insulin sensitivity, enhances peripheral glucose uptake, increases fatty acid oxidation, and decreases absorption of glucose from the gastrointestinal tract, as evident in ....Metformin increases insulin sensitivity and basal glucose clearance in type 2 (non-insulin-dependent) diabetes mellitus. Aust N Z J Med. 1991 Oct;21(5):714-9.
Response: Thank you for pointing out the error in the write-up. Indeed, Metformin and Thiazolidinediones improve insulin sensitivity. We have made suitable changes in the write-up from lines 68-73.
Comment 2: Lean NAFLD should be more precisely defined evaluating the abdominal adiposity, datum not always present when dealing with this type of NAFLD, as evident in....Lean-non-alcoholic fatty liver disease increases risk for metabolic disorders in a normal weight Chinese population. World J Gastroenterol. 2014 Dec 21;20(47):17932-40.
Response: Thank you for raising this point. There remain many uncertainties about the mechanistic basis for lean NALFD. As this discussion may not enrich the premise for this review, we have decided to remove this mention from the introduction section.
Comment 3: Authors correctly state that ....However, these strategies focus on alleviating the effect of NAFLD rather that resolving the pathogenesis of the disease...but they should quote the pertinent reference concerning the fact that the inner mechanisms of NAFLD are far from being clarified that is....J. Clin. Med. 2020, 9(1), 15; https://doi.org/10.3390/jcm9010015.
Response: Thank you for the suggestion. We have made suggested amendments in the write-up from lines 76-78.
Comment 4: Authors state that ...Moreover, literature also reports upregulation of VEGF during early stages of NAFLD such as steatosis...
But, one of the first reports on the role of this main cytokine in human NAFLD was .....Could inflammatory markers help diagnose non-alcoholic steatohepatitis? Eur J Gastroenterol Hepatol. 2009 May;21(5):504-11.
Response: The reference suggested is a valuable piece of evidence to claim that VEGF is activated in NAFLD progression and holds great potential in NAFLD diagnosis. We have incorporated it in our manuscript in lines 372-375.
Comment 5: Authors should state that most parts of the studies that they present are performed using animal models of NAFLD, but they do not completely mirror human NAFLD as evident in... Animal Models of Non-alcoholic Fatty Liver Disease-A Starter's Guide. Nutrients. 2017 Sep 27;9(10):1072. doi: 10.3390/nu9101072. PMID: 28953222; PMCID: PMC5691689.
Response: Thank you for bringing this important point to our notice. We have added this point in the conclusion section. Wherein we discussed that although there are various animal models, each model comes with a few limitations to fall short of actually mimicking the pathology of NAFLD in humans. We also discuss that animal models may not predict the toxic effects of the drugs in humans. (line 537-544)
Comment 6: The long-term effects concerning the use of these new molecules should be carefully followed and ascertained, as well as the possible drug-drug interactions, as evident in...Molecular Interactions between NAFLD and Xenobiotic Metabolism. Front Genet. 2013 Jan 22;4:2.
Response: We appreciate this suggestion and understand that discussing challenges caused in drug metabolism in NAFLD patients is a grave concern and it must be addressed. We discuss this issue in lines 544-552.
Comment 7: These last points should be at large discussed to offer readers a broader view of the topic.
Response: We agree with this suggestion and thus we have elaborated on the above-mentioned points in the conclusion section.
Reviewer 3 Report
Complete and well conducted review on receptor tyrosine kinases in the management of non-alcoholic fatty liver disease
Author Response
We thank you for your encouraging comments. Here is a response to the comment from the reviewer.
Comment 1: Complete and well conducted review on receptor tyrosine kinases in the management of non-alcoholic fatty liver disease
Response: Thank you very much for your word of support.
Round 2
Reviewer 2 Report
Manuscript improved according to comments